# Exploring the Social Determinants of Mental Health in Colombian Young Adults

**DOI:** 10.3390/ejihpe15070133

**Published:** 2025-07-13

**Authors:** Mónica Roncancio-Moreno, Rita Patricia Ocampo-Cepeda, Arcadio de Jesús Cardona-Isaza

**Affiliations:** Department of Psychosocial Studies, Faculty of Psychology, Universidad del Valle, Cali 760042, Colombia; rita.ocampo@correounivalle.edu.co (R.P.O.-C.); arcadio.cardona@correounivalle.edu.co (A.d.J.C.-I.)

**Keywords:** young adults, mental health, social determinants of health, prevalence, violence

## Abstract

Background: The prevalence of mental health problems among young adults has increased exponentially worldwide, with significant implications for public health. This study explores the structural and intermediate social determinants of health associated with emotional well-being and distress in Colombian young adults. Methods: A cross-sectional study was conducted with 1232 university students aged 18–28 years residing in Cali, Colombia. The Social Determinants of Mental Health Questionnaire for Young Adults (SDMH) was administered to assess structural and intermediate determinants. Participants were categorized into two groups: those with mental health diagnoses (n = 252) and those without (n = 980). Descriptive, comparative association and predictive analyses were conducted. Results: Significant differences were found between groups, with diagnosed participants showing higher exposure to violence and suicidal ideation. Women with diagnoses predominantly presented anxiety and depression, while men showed more externalizing symptoms. Violence experiences emerged as the strongest predictors of emotional distress in both groups. Psychosocial life skills and perceived social support were the most robust protective factors for emotional well-being, regardless of diagnostic status. Conclusions: This study reveals that multiple risk factors accumulate rather than single extreme factors characterizing mental health vulnerability. Violence prevention, life skills development, and social support enhancement should be prioritized in public health interventions for young adults. The findings emphasize the need for multisectoral approaches addressing social determinants to promote youth mental health.

## 1. Introduction

In recent years, the prevalence of mental health problems and disorders among young adults has increased exponentially, as indicated by recent reports made by different international organizations such as the World Health Organization (WHO), Sapien Labs, and the Panamerican Health Organization (PHO), among others. According to information provided by [28] ([28]), mental health problems significantly affect young adults worldwide, especially in relation to unwanted, strange and obsessive thoughts, guilt and blame, mood swings, feelings of sadness or hopelessness, fear and anxiety, and feelings of detachment from reality. In addition to these symptoms, recent data indicate a significant—and progressive—deterioration in the mental health and well-being of young adults over the past two decades, particularly in relation to measures of anxiety and depression, life satisfaction, and both negative and positive affect ([28]).

Likewise, the WHO reports that suicide is the third leading cause of death in older adolescents and young adults worldwide and identifies anxiety disorders as the most prevalent in this age group ([34]). In the same vein, a recent study, considered significant for its implications for the global mental health landscape of young people, was developed by [20] ([20]). In this research, mental health problems in young people under the age of 25 were tracked (longitudinal data between 1996 and 2023 in several countries) and a substantial increase in anxiety, psychological distress, self-harm, suicide and depressive symptoms was found; these results are referred to by the authors as a “global youth mental health crisis” ([20]).

These epidemiological data highlight the mental health of young people as a global public health problem, which is why academia is called upon to propose research to characterize and understand how our young adults are facing these challenges. This article presents the results of a mental health questionnaire administered to young adults and analyzes the prevalence of these problems from the social-determinants-of-health perspective applied in Cali, Colombia.

### Social Determinants of Health and Their Role in Mental Health

From the WHO perspective, [4] ([4]) characterize social determinants as the broad spectrum of social, political, economic, environmental, and cultural variables that profoundly influence health outcomes at the individual level. In turn, these determinants are divided into structural and intermediate ones. Structural determinants are those that produce social stratification, such as income, education, gender, race and sexuality. On the other hand, intermediate determinants underlie social stratification and shape differences in exposure and vulnerability to factors that affect health, for example, labor and living conditions, housing and access to healthcare.

Regarding the social determinants of mental health, [4] ([4]) point out that social support, community networks, adverse childhood experiences, and the interrelation between discrimination and stigma contribute to the detriment of the population’s mental health. Likewise, they state that a multisectoral approach is necessary to cover the diversity of determinants that influence mental health.

In addition to social determinants, it could be considered that the vital moment of young adulthood makes individuals especially vulnerable to experiencing mental health challenges, as confirmed by several studies conducted within this age group ([3]; [6]; [12]; [16]). Moreover, considering the social determinants of health alongside the moment of the vital development of young people, we can affirm that this intricate network of social, economic, environmental and personal factors profoundly influences their mental health outcomes ([19]).

Key elements of young people’s life course related to their mental health have been documented; for example, poor working conditions and limited access to higher education increase the risk of anxiety and depression. Similarly, the pursuit of economic independence related to educational debt burden or low entry-level wages makes young people susceptible to chronic stress ([32]).

Regarding specific data at the Latin American level, [10] ([10]) surveyed young adults in Colombia, Argentina, and Peru to inquire about factors associated with symptoms of anxiety and depression in this population. The results showed that female sex, stressful life events in the last year and in previous years, substance use, artistic activities, and the problematic use of social networks were associated with higher probabilities of depression and anxiety, while participation in sports activities was linked to better mental health indicators. The data provided by this latest research highlight the urgency of exploring in depth the deteriorating mental health of young adults and the development of measures in terms of public policies that can counter the advancement and expansion of this problem.

In a recent analysis of Colombia’s health situation, data from 2019 to 2023 reported the prevalence of mental disorders in the general population ([1]). The results reveal a higher prevalence of depression and anxiety among individuals aged 15 to 29 compared to other age groups. Regarding suicide attempts, it was found that young people aged 15 to 26 accounted for the highest percentage of cases during the analyzed period (51.5%). By the five-year intervals, it was found that between 10 and 29 years of age, a great majority of suicide attempts were recorded (72.9%). In terms of suicide deaths, the highest percentage of cases occurred in young people and adults (52.3%). By the five-year age intervals, the greatest proportion of cases occurred between ages 15 and 34 (47%).

Thus, in the face of the crisis posed by the undermining of youth mental health, it is necessary to promote an academic debate that focuses on thinking about this problem as a public health situation that has a long-term impact on the development and overall well-being of young people. Studying the mental health of young adults is a priority task for both the research agenda and public policy, which is why this study aims to explore the structural and intermediate determinants that are associated with and influence emotional well-being and distress in young adults with and without a mental health diagnosis in the city of Cali using the Social Determinants of Mental Health Questionnaire for Young Adults (SDMH) from [26] ([26]).

## 2. Method

This study aimed to explore the structural and intermediate determinants associated with and influencing emotional well-being and distress in young adults, both with and without a mental health diagnosis.

### 2.1. Participant Characteristics

The final sample comprised 1232 young university students residing in Cali, Colombia, aged 18 to 28 years (M = 20.48, SD = 2.26), of whom 46.8% (n = 576) were women. Of these, 980 participants (M = 20.48, SD = 2.29) reported no history of mental health diagnosis; 519 were men (53%) and 419 were women (47%). The remaining 252 participants (M = 20.51, SD = 2.19) reported a mental health diagnosis, which included 137 men (54.4%) and 115 women (45.6%). The participants were from a low socioeconomic status group. Additional participants characteristics are detailed in Table 1. Participants met the eligibility criteria if they satisfied the following conditions: (1) aged 18 to 28 years (2) and provided signed informed consent. Individuals outside this age range or those who did not provide consent were excluded. We performed a sample calculation based on 4,754,451 people, which is the current projection of the National Administrative Department of Statistics ([7]) of persons in the country between 18 and 28 years of age. The adjusted sample required a 15% loss to be representative, constituting 412 people, using a confidence level of 1 − α = 0.99, a precision of d = 0.03 and an expected proportion of *p* = 0.05. The sample of participants exceeds the projection. In addition, the ethnic representation of mestizos, afro-descendants and indigenous people in the study is in line with the national ethnic distribution for indigenous people, which is 5%, and afro-descendants, which is higher at 7.8%; this is due to the cultural richness, population mobility, and confluence of ethnic communities in the city.

### 2.2. Instrument

The *Social Determinants of Mental Health Questionnaire for Young Adults* (SDMH; [26]) was used to assess the social determinants of mental health. This instrument was developed based on the social-determinants-of-health model proposed by the World Health Organization ([33]).

The SDMH consists of 72 items that explore both structural and intermediate determinants. It is a mixed-format questionnaire, including open-ended, dichotomous, multiple-choice, and Likert-type questions. It facilitates the identification of risk and protective factors, as well as the evaluation of emotional well-being and distress. In addition, it includes six outcome variables that capture physical, emotional and psychological well-being and distress experienced over the past six months and the past week.

The structural determinants consist of sixteen conditions, encompassing aspects such as socioeconomic conditions, access to health services, and ethnic, educational, and work characteristics. Intermediate determinants encompass twenty-one factors grouped into 5 dimensions related to behavioral and psychosocial aspects, including physical and emotional symptoms, risk behaviors (such as substance use and suicidal behavior), healthy lifestyle habits, psychosocial life skills, and perceived social support.

The instrument has a Confirmatory Factor Analysis (CFA), which has already been published and has the current sample data. The fit indices were adequate for the dimensions evaluated with values of the Comparative Fit Index (CFI) between 0.87 and 0.94 and Tucker–Lewis index (TLI) between 0.85 and 0.94. For the total scale, the values were CFI = 0.82, TLI = 0.82, and RMSEA = 0.042, CI 90% (0.041, 0.043). In addition, the instrument showed configural, metric, scalar, structural and residual invariance by gender ([26]).

The questionnaire can be administered individually or collectively, in either digital or paper format. It was designed with clear and inclusive language incorporating cultural and contextual adaptations for Latin American populations. The structure of the survey and the reliability of the factors can be found in Appendix A.

### 2.3. Procedure

The data collection phase took place from August to November 2023, involving 1232 university students aged 18 to 28, all residing in Cali and neighboring cities in Colombia. The selection method employed was convenience sampling. The administration of the questionnaires was conducted through digital platforms, forming small working groups and relying on the guidance of licensed psychologists with updated professional registration. Voluntary participation was documented through informed consent signed by each student. At the end of the digital administration of the questionnaire, participants automatically received the university’s care pathways and local health services for various types of problems, including suicidal behavior and exposure to violence. Contact information for free mental health services and 24 h helplines available in the region was also provided. Likewise, the application was conducted in person with the intention that psychologists could attend to participants’ questions, offer guidance on care services, and activate pathways when necessary; the indicated protocols were validated and active in local health services.

### 2.4. Analytic Strategy

The data were analyzed using IBM SPSS Statistics (version 25; [14]). Descriptive statistics were first computed to characterize the participants and explore differences between those who reported a mental health diagnosis and those who did not. Subsequently, Mann–Whitney U tests were conducted to compare the groups on intermediate determinants, as several variables did not meet the assumptions of normality. Additionally, Spearman’s rank-order correlations were used to examine associations between intermediate risk and protective factors, distinguishing between participants with and without a mental health diagnosis.

Finally, hierarchical regression analyses were performed to evaluate the effects of structural and intermediate determinants on two dependent variables: negative emotional symptoms and emotional well-being experienced over the past six months. Models were estimated separately for both groups (with and without a diagnosis) and structured in two steps. Structural variables were entered as control variables in the first block, followed by intermediate factors in the second. Details on variable coding are available in Appendix A.

## 3. Results

### 3.1. Descriptive Results

The descriptive results show differences in the proportion of participants with and without a mental health diagnosis. Among those with a diagnosis (n = 252), a higher proportion reported suicidal ideation (68.7% vs. 43.1%) compared to those without a diagnosis (n = 980). This group also reported greater exposure to stressful situations (87.7% vs. 79.3%), more chronic health problems (20.6% vs. 11.7%), and a higher rate of low or very low satisfaction with their studies (34.9% vs. 7.4%). Additionally, a larger proportion of participants with a diagnosis reported having experienced physical (18.7% vs. 11.9%), sexual (12.7% vs. 6.7%), and psychological violence (28.6% vs. 16.5%). While most participants in both groups were affiliated with the healthcare system, those with a diagnosis reported greater difficulties related to both structural and intermediate determinants of mental health. The descriptive results are detailed in Table 1. Additionally, Table 2 presents the descriptive statistics and reliability data of the intermediate determinants assessed with the Social Determinants of Mental Health Questionnaire for Young Adults.

#### Mental Health Diagnosis

In the data analysis from 252 individuals with a mental health diagnosis, various disorders were identified, with the most common being anxiety and depression-related disorders, followed by eating disorders, neurodevelopmental disorders, mood disorders, and psychotic disorders.

The most prevalent diagnosis was anxiety, which included various forms such as generalized anxiety disorder, social anxiety disorder, panic attacks, and emotional crises. This group accounted for 112 cases, representing 44.4% of the sample, with an average onset age of 18.6 years. Depression followed, with 41 individuals diagnosed (16.3% of the total) and an average onset age of 18.4 years. In many cases, both disorders co-occurred in the form of mixed anxiety–depressive disorder, which was reported by 28 individuals (11.1%), with an average onset age of 18.9 years. Eating disorders (EDs), such as anorexia and bulimia, were identified in 15 participants (6.0%), with an average diagnosis age of 19.1 years, indicating they tend to emerge during the transition from adolescence to adulthood. In contrast, Attention-Deficit/Hyperactivity Disorder (ADHD) was less common (six cases, 2.4%) but had a much earlier average onset age of 10.5 years, consistent with its typical detection in school settings.

Borderline Personality Disorder (BPD) was diagnosed in 10 participants (4.0%) with an average onset age of 18.0 years, while Obsessive–Compulsive Disorder (OCD) was reported in 6 individuals (2.4%), with an average diagnosis age of 17.5 years. Stress-related disorders, including chronic stress and Post-Traumatic Stress Disorder (PTSD), were reported by nine participants (3.6%), with an average onset age of 18.3 years. Psychotic disorders, such as schizophrenia or psychosis-related experiences, were identified in four cases (1.6%) and had an average onset age of 17.1 years. Notably, 40.1% (approximately 101 individuals) reported more than one mental health diagnosis, indicating a high prevalence of comorbidity—most commonly the co-occurrence of anxiety and depression.

Regarding sex differences, both men (n = 137) and women (n = 115) displayed the most common mental health diagnoses, namely anxiety and depression. However, the analysis revealed notable differences in their frequency and type. Anxiety was the most common diagnosis for both sexes, with 56 cases in women (48.7%) and 56 in men (40.9%). Depression showed similar prevalence across groups, with 20 cases in each corresponding to 17.4% of women and 14.6% of men.

Nonetheless, eating disorders (EDs) were more prevalent among women (5.2%) than men (1.5%), as was Borderline Personality Disorder, which also showed a higher prevalence in women (2.6% versus 0.7%). In contrast, men more frequently reported symptoms related to stress, anger management, and behavioral issues. Overall, 40.1% of participants had multiple diagnoses, with the most frequent combinations involving anxiety and depression—reported 45 and 47 times, respectively. The average age at diagnosis was similar across sexes: 18.6 years for anxiety, 18.4 years for depression, and 18.9 years for mixed anxiety–depressive disorder.

### 3.2. Comparative Results

The comparative analysis of participants with and without a mental health diagnosis detected several statistically significant differences across various psychosocial and well-being variables. When assessing the magnitude of these differences through effect sizes (r), most fall within the small range (r between 0.10 and 0.29). For example, domains such as well-being at home (r = 0.132), well-being in religious organizations (r = 0.111), and expectation of social support (r = 0.082) showed statistically significant differences with small effect sizes. This suggests that participants without a diagnosis generally reported higher levels in these areas. In contrast, suicidal behavior, emotional symptoms, and physical symptoms, both in the past week and over the last six months, not only reached statistical significance but also presented small to moderate effect sizes (e.g., suicidal behavior with r = 0.27, emotional symptoms in the last week with r = 0.17). These findings highlight more substantial differences between the groups, suggesting that mental health diagnoses are more clearly associated with increased emotional distress and suicide-related indicators. Although several statistically significant differences were found—particularly in emotional symptoms and suicidal behavior—the unequal group sizes (252 with a diagnosis vs. 980 without) may have influenced the results. The Mann–Whitney U test is sensitive to sample imbalance, which can increase the likelihood of significance and reduce effect sizes. Therefore, results with small effects should be interpreted with caution. The results of the difference analyses are presented in Table 3.

### 3.3. Association Results

Spearman correlations indicated significant associations between intermediate determinants in both groups, although the strength and pattern of these relationships varied based on mental health status. Among participants without a mental health diagnosis, correlations were more extensive but weaker. Protective factors such as well-being at home, the expectation of social support, and healthy lifestyle habits were positively associated with various dimensions of well-being and negatively correlated with emotional distress and suicidal behavior. For instance, emotional distress showed a moderate negative correlation with psychosocial skills for life (r = −0.28) and healthy habits (r = −0.26).

In contrast, participants with a diagnosis showed fewer but stronger correlations, particularly regarding psychological distress. Emotional symptoms were more strongly associated with suicidal behavior (r = 0.42), and emotional distress showed stronger negative correlations with well-being at home (r = −0.35) and healthy habits (r = −0.29). Stronger associations were also found between psychosocial skills and various indicators of well-being, such as healthy habits (r = 0.44) and well-being in the last *week* (r = 0.41). While protective and risk factors were associated in both groups, these relationships were more focused and intense among participants with a mental health diagnosis, particularly around emotional distress and suicidal behavior. The detailed correlation results between intermediate determinants for each group are provided in Appendix A.

### 3.4. Predictive Findings

A hierarchical regression analysis was performed using a sample of 980 participants without a mental health diagnosis, aiming to predict negative emotional symptoms and psychological distress experienced over the past six months, based on well-being across different environments and various risk and protective factors (Table 4). In the first step, structural variables were included, and the model was statistically significant, explaining 13% of the variance (R^2^ = 0.13; F (_19, 961_) = 7.333, *p* < 0.001). Variables strongly associated with emotional symptoms included being in a romantic relationship (β = −0.27, *t* = −2.02, *p* = 0.006, 95% CI [−17.37, −0.28]), which was linked to lower emotional symptomatology; lower satisfaction with academic life (β = −0.14, *t* = −4.63, *p* < 0.001, 95% CI [−1.52, −0.61]); and having experienced physical (β = 0.17, *t* = 5.46, *p* < 0.001, 95% CI [2.78, 5.89]), sexual (β = 0.13, *t* = 4.27, *p* < 0.001, 95% CI [2.34, 6.31]), and psychological violence (β = 0.24, *t* = 7.65, *p* < 0.001, 95% CI [3.94, 6.66]), all of which were associated with higher levels of emotional symptoms. Additionally, a significant relationship was found between emotional symptoms and having experienced aggression on social media (β = 0.09, *t* = 2.87, *p* = 0.011, 95% CI [1.85, 9.80]).

In the second step, intermediate factors were included, enhancing the model’s explanatory power, accounting for 29% of the variance (R^2^ = 0.29; F (_34, 961_) = 11.174, *p* < 0.001), with a significant increase of ΔR^2^ = 0.16. The variables that contributed most to this explanation were lower levels of healthy lifestyle habits (β = −0.15, *t* = –4.45, *p* < 0.001, 95% CI [−0.30, −0.12]), higher levels of worry and emotional distress (β = 0.21, t = 6.69, *p* < 0.001, 95% CI [0.34, 0.62]), greater suicidal behavior (β = 0.20, *t* = 6.53, *p* < 0.001, 95% CI [1.20, 2.23]), and problems due to problematic internet use (β = 0.15, *t* = 4.47, *p* < 0.001, 95% CI [0.24, 0.61]). Other variables, such as drug use (β = 0.06, *t* = 1.98, *p* = 0.048, 95% CI [0.002, 0.41]) and related conflicts (β = −0.07, *t* = −1.94, *p* = 0.052, 95% CI [−0.32, 0.00]), revealed marginal yet notable effects. The full results of all reported regression analyses can be found in Appendix A.

A second hierarchical regression analysis was conducted to predict emotional symptoms and psychological distress experienced over the past six months by 252 participants with a current mental health diagnosis. In the first step, structural variables were included, and the model was statistically significant, explaining 20% of the variance (R^2^ = 0.20; F (_19, 250_) = 3.058, *p* < 0.001). Among the variables strongly associated with higher emotional symptoms included having experienced physical violence (β = 0.29, *t* = 4.49, *p* < 0.001, 95% CI [3.56, 9.12]) and psychological violence (β = 0.26, *t* = 3.88, *p* < 0.001, 95% CI [2.35, 7.20]).

In the second step, intermediate factors were added, significantly increasing the explained variance to 38% (R^2^ = 0.38; F (_34, 961_) = 3.874, *p* < 0.001), with a ΔR^2^ increase of 0.18. Worry and emotional distress (β = 0.28, *t* = 4.37, *p* < 0.001, 95% CI [0.35, 0.92]) along with problems due to problematic internet use (β = 0.18, *t* = 2.65, *p* = 0.009, 95% CI [0.14, 0.96]) were key contributors to the model.

A third hierarchical regression analysis was performed using a sample of 980 participants without a mental health diagnosis, aiming to predict emotional well-being over the past six months based on structural and intermediate variables. In the first step, structural variables were included, and although the model was statistically significant, it showed low explanatory power, accounting for 5% of the variance (R^2^ = 0.05; F (_19, 961_) = 1.391, *p* < 0.001). Within this block, the only variables that showed significant associations were ethnic group (β = −0.08, *t* = −2.31, *p* = 0.021, 95% CI [−1.18, −0.10]), indicating lower well-being among participants belonging to ethnic minority groups, and satisfaction with academic life (β = 0.10, *t* = 3.39, *p* < 0.001, 95% CI [0.16, 0.60]), which was positively associated with well-being.

In the second step, intermediate factors were included, and the model’s explanatory capacity improved, reaching 13% of the variance (R^2^ = 0.13; F (_34, 961_) = 2.990, *p* < 0.001), with a significant increase of ΔR^2^ = 0.08. The variables that contributed most significantly to the prediction of well-being were psychosocial life skills (β = 0.22, *t* = 6.28, *p* < 0.001, 95% CI [0.08, 0.14]), expected social support (β = 0.08, *t* = 2.06, *p* = 0.039, 95% CI [0.002, 0.06]), access to sociocultural and recreational well-being (β = −0.09, *t* = −2.37, *p* = 0.018, 95% CI [−0.07, −0.01]), and lower well-being at work (β = −0.12, *t* = −2.04, *p* = 0.042, 95% CI [−0.11, −0.002]).

Finally, a hierarchical regression analysis was conducted to predict emotional well-being over the past six months in a sample of 252 participants with a current mental health diagnosis. In the first step, structural variables were included. The model explained 10% of the variance (R^2^ = 0.10; F (_19, 250_) = 1.391, *p* = 0.132), although it was not statistically significant. The only variables that showed significant associations with higher well-being were satisfaction with academic life (β = 0.20, *t* = 3.00, *p* = 0.003, 95% CI [0.22, 1.07]) and affiliation with healthcare services (β = 0.14, *t* = 2.07, *p* = 0.039, 95% CI [0.13, 5.04]). Other structural variables did not show statistically significant effects.

In the second step, intermediate factors were incorporated, significantly improving the model, which explained 32% of the variance (R^2^ = 0.32; F (_34, 961_) = 2.990, *p* < 0.001), with an increase of ΔR^2^ = 0.22. Variables significantly associated with higher levels of well-being included well-being at home (β = 0.023, *t* = 3.31, *p* < 0.001, 95% CI [0.09, 0.36]), well-being at work (β = 0.34, *t* = 2.49, *p* = 0.013, 95% CI [0.03, 0.29]), and psychosocial life skills (β = 0.23, *t* = 3.29, *p* < 0.001, 95% CI [0.04, 0.17]). On the other hand, suicidal behavior was negatively associated with well-being (β = −0.18, *t* = −2.65, *p* = 0.009, 95% CI [−1.03, −0.15]), as were problems due to problematic internet use (β = −0.26, *t* = −3.58, *p* < 0.001, 95% CI [0.16, 0.56]), indicating an adverse impact.

Across the regression analyses, emotional symptoms were mainly predicted by exposure to violence, worry, suicidal behavior, and problematic internet use. In contrast, emotional well-being was consistently associated with psychosocial skills, well-being in daily environments, and social support.

## 4. Discussion

This study aimed to examine the structural and intermediate social determinants of health associated with and influencing emotional well-being and distress in young adults with and without mental health diagnoses. To achieve this objective, the Social Determinants of Mental Health Questionnaire for Young Adults (SDMH; [26]) was administered.

The results confirm the existence of differences between individuals with and without a diagnosis. In both groups, common risk and protective factors with high prevalence are identified, particularly those related to experiences of violence and lack of psychosocial support. For example, risk scores stand out in suicide attempts, exposure to stressful situations, and violence. These results align with recent studies by [20] ([20]) and highlight the scale of a problem that transcends local contexts to a global level.

In the descriptive analyses, participants were classified into two cohorts: those with mental health diagnoses and those without diagnoses. In the group with diagnoses, women presented emotional disorders such as anxiety (48.7%) and depression (17.4%) more frequently, while in men, a higher prevalence of symptoms related to stress, anger management problems, and problematic behaviors were identified. This pattern is consistent with the previous literature that points to gender differences in symptomatic presentation, with greater internalizing expression in women and externalizing in men ([17]). Previous studies have identified that female sex, substance use, stressful life events, and artistic activities are significantly associated with anxious and depressive symptomatology in young adults ([13]; [25]), which is consistent with the findings of this study.

In both groups, with a diagnosis and without a diagnosis, high exposure to adverse events is observed, especially physical, sexual, and psychological violence. In line with this trend, it is concerning that almost half of young people without a formal diagnosis (43%) and with a diagnosis (68%) experience suicidal ideation, which directly relates to the results of [1] ([1]), which indicates that individuals aged 15 to 29 years present the highest prevalence for depression and anxiety and concentrate more than 50% of suicide attempts. These data could also be interpreted as a gap in timely diagnosis and treatment aggravated by stigma, social exclusion, and discrimination, which has been primarily documented in Latin American countries ([34]).

While suicidal ideation raises alarms, the high exposure of young people to stressful situations (87.7% in people with a diagnosis and 79.3% without a diagnosis) and the prevalence of experiences of violence in all its forms is no less important. The convergence of structural and intermediate determinants—such as violence, academic dissatisfaction, emotional distress, and suicidal behaviors—creates a highly complex landscape that jeopardizes the present and future well-being of participants.

Specifically concerning forms of violence, this study documented physical, sexual, psychological violence, violence occurring on social networks, bullying, and forced displacement. The data also indicate that participants in the diagnosed group have been exposed to more adverse events than those in the group without a diagnosis. These data correspond to research by [18] ([18]), [29] ([29]), and [15] ([15]), reinforcing evidence that early adversity, especially socioeconomic challenges and family conflicts, has lasting effects on mental health. However, the study by [15] ([15]) highlights that the type and configuration of adversities is more important than total number, underscoring the need for targeted interventions tailored to specific adversity types.

On the other hand, the results suggest that significant differences exist between participants with and without a diagnosis, though their magnitude is not particularly large. When comparing groups, the most notable differences stem less from the magnitude of effects and more from the accumulation of risk factors among diagnosed individuals. The most pronounced significant differences were observed in variables such as well-being at home, the expectation of social support, and almost notably, emotional and physical symptoms and suicidal behavior. Effect sizes ranged from small values between 0.17 and 0.27. This suggests that young people with diagnoses present higher levels of accumulated distress and greater exposure to adverse intermediate determinants. It is important to highlight the role of indicators such as satisfaction with academic life and the perception of social support as factors that can modulate these differences. This accumulation of risk factors, rather than the presence of a single factor with an extremely large effect, seems to be what characterizes the group with mental health diagnosis. Similar findings have been reported by [8] ([8]), [19] ([19]), and [22] ([22]), among others.

The results derived from the analysis reveal that, in predictive terms, certain determinants exert a stronger impact on emotional distress. Among individuals without a diagnosis, being single, experiencing chronic health problems, and feeling dissatisfied with studies were significantly associated with greater distress. These results align with contemporary evidence, in which it was also found that these factors function as predictors of psychological deterioration in adolescents and young adults ([24]). In both groups, experiences of violence—physical, sexual, and psychological—were the strongest predictors of negative emotional symptoms, with higher effects as the quantity and intensity of reported forms of violence increase. Similarly, studies by [5] ([5]) and [11] ([11]) confirm that exposure to multiple forms of violence predicts more severe outcomes than other stressors. It was also identified that unhealthy lifestyle habits, problematic internet use, drug consumption, and emotional stress contribute to distress, depressive symptomatology, and suicidal ideation ([9]), which points to the need to design preventive interventions that address these factors in an integrated manner.

A central finding of this study is that, regardless of whether young people have or do not have a mental health diagnosis, the most robust and constant protective factors in improving well-being were life skills and perceived social support. In the group without a diagnosis, the development of psychosocial life skills and expectation of social support significantly predicted higher levels of well-being. In the group with a diagnosis, these variables also showed relevant positive effects, along with satisfaction with academic life, well-being at home and at work. Similar results were found by [23] ([23]), [30] ([30]), [27] ([27]), among others. This underscores the urgency of implementing institutional programs that strengthen these skills, reduce barriers to service access, and promote stronger support networks.

In conclusion, this study allows the identification of differential, common, and predictive factors of distress and emotional well-being in Colombian young adults from a social-determinants-of-health approach. Similarly, these data reflect the profound impact that forms of violence have on young people’s mental health and coincide with international studies that indicate interpersonal violence as a significant predictor of psychological distress and suicidal risk in this population.

Moreover, the need for multisectoral interventions oriented towards the prevention of all forms of violence is clearly a public health priority. The findings highlight the urgency of developing public policies and institutional interventions focused on violence prevention, strengthening life skills, and improving social support as fundamental pillars in promoting youth mental health. Multiple studies have shown that interventions based on the development of psychosocial skills significantly reduce violence and improve social behavior ([31]) and are associated with reductions in aggression and improvements in the mental health of adolescents and youth, including those exposed to adversities ([21]).

Several important limitations should be acknowledged in this study. First, self-reported lifetime mental health diagnoses may introduce classification biases through under-reporting or over-reporting, potentially affecting the accuracy of between-group comparisons. Second, while the statistical analyses based on hierarchical regressions and non-parametric comparisons provide valuable information indicative of factors affecting the participant population, they do not adequately control for confounding variables. Future research would benefit from implementing more robust statistical approaches, including longitudinal data collection, precise group matching techniques, and structural equation modeling (SEM) to reduce Type I error risk. Given these methodological considerations, this study should be regarded as exploratory in nature. The cross-sectional design inherently limits the ability to establish causal relationships and determine directionality of effects between factors. Future longitudinal studies incorporating data from multiple sources and instruments could build upon the preliminary observations regarding social determinants of mental health presented here, providing a more comprehensive understanding of these complex relationships.

Another limitation was conducting comparisons between groups with unequal sizes using a non-parametric test. This decision responded to the exploratory nature of the study and the need to preserve the complete sample, which resulted in a lack of systematic control for confounding variables. For future studies, strict comparative analyses using techniques such as propensity score matching (PSM) or multivariate models that improve the internal validity of comparisons are recommended ([2]).

Finally, the relative homogeneity of the sample’s socioeconomic conditions constitutes a limitation of this study, as most participants came from economically vulnerable backgrounds. Future research should include more heterogeneous samples to provide deeper insights into the impact of structural conditions on mental health. It is suggested that future research incorporates this dimension and uses probabilistic sampling strategies to improve representativeness. Additionally, it would be important to expand the sample to other regions of the country to contrast the results obtained in Cali with other sociocultural realities.

Future studies should prioritize longitudinal follow-ups that allow the examination of how psychosocial life skills and perceived social support evolve throughout the life trajectory until job insertion. Similarly, future studies could be developed in earlier phases of development such as preadolescence.

## Figures and Tables

**Table 1 ejihpe-15-00133-t001:** Descriptive statistics of the sociodemographic aspects and structural determinants.

Variable	Items	Participants with a Diagnosis (n = 252)		Participants Without a Diagnosis (n = 980)	
Sociodemographic data					
		n	%	n	%
Ethnicity	Mestizo	196	77.8	699	71.3
	Afro descendant	42	16.7	214	21.8
	Indigenous	14	5.6	65	6.6
	Roma	196	77.8	2	0.2
Structural determinants associated with the vital transition					
Family nucleus	Family of origin	239	94.8	937	95.6
	Other relatives	9	3.6	31	3.2
	Other people, not relatives	2	0.8	4	0.4
	Couple and children	1	0.4	3	0.3
	Single couple	1	0.4	1	0.1
	Alone	239	94.8	4	0.4
Marital status	Single	236	93.7	917	93.6
	Married	5	2.0	7	0.7
	Free union	11	4.4	53	5.4
	Divorced			2	0.2
	Widower			1	0.1
Study	Yes	11	4.4	980	100
Satisfaction with studies	Do not know	5	2.0	34	3.5
	Very low	25	9.9	15	1.5
	Low	63	25.0	58	5.9
	Medium	107	42.5	263	26.8
	High	41	16.3	456	46.5
	Very high	11	4.4	154	15.7
Works	Yes	82	32.5	301	30.7
Dependent persons	Yes	24	9.5	75	7.7
Affiliated with health services	Yes	241	95.6	938	95.7
Intermediate determinants					
Chronic health problem	Yes	52	20.6	77	11.7
Treatment effectiveness: health problems	Yes	40	15.9	60	6.1
Functional diversity	Yes	44	17.5	132	20.1
Treatment or rehabilitation	Yes	23	9.1	108	11.0
Exposure to stressful situations	Yes	473	87.7	777	79.3
	Many years ago	120	47.6	401	40.9
	More than six months ago	56	22.2	191	19.5
	In the last six months	26	10.3	122	12.4
	Currently	19	7.5	63	6.4
Victim of violence					
	Physical	47	18.7	117	11.9
	Sexual	32	12.7	66	6.7
	Psychological	72	28.6	162	16.5
	In social networks	3	1.2	15	1.5
	Bullying	22	8.7	80	8.2
	Forced displacement	3	1.2	37	3.8
Suicidal behavior	Suicidal ideation	173	68.7	422	43.1

**Table 2 ejihpe-15-00133-t002:** Descriptive statistics and internal consistency of subscales measuring intermediate determinants of mental health in youth with and without a mental health diagnosis (N = 1232).

			With Diagnosis (n = 252)			Without Diagnosis (n = 980)		
Variables	Items	Scale Score	M	SD	α	M	SD	α
Well-being at home	4	0–6	16.54	4.00	0.74	17.74	3.63	0.69
Well-being in the neighborhood	4	0–6	12.86	5.15	0.59	14.00	4.83	0.59
Well-being in religious organizations	4	0–6	4.96	6.69	0.84	7.04	7.81	0.87
Social welfare, culture and recreation	4	0–6	8.90	8.75	0.90	9.34	8.73	0.90
School wellness	4	0–6	16.91	3.57	0.40	17.02	3.59	0.38
Well-being at work	4	0–6	6.21	7.97	0.91	6.47	8.13	0.92
Expectation of social support	8	0–6	24.65	8.85	0.65	26.47	9.16	0.68
Healthy lifestyle habits	8	1–6	25.09	6.41	0.75	25.50	5.90	0.70
Psychosocial skills for life	10	1–5	34.77	8.37	0.90	36.66	7.72	0.90
Worry and emotional distress	5	0–4	11.54	3.67	0.69	10.45	3.60	0.73
Suicidal behavior	3	0–1	1.61	1.17	0.71	0.85	0.95	0.62
Drug use	5	0–4	3.18	3.02	0.62	2.63	2.52	0.63
Conflicts due to drug use	7	0–2	4.71	3.64	0.92	4.52	3.52	0.94
Problematic internet use	4	0–4	6.58	2.72	0.48	6.50	2.71	0.45
Problems due to problematic internet use	7	0–2	6.49	2.74	0.83	6.19	2.90	0.85
Emotional symptoms in the last week	8	1–5	26.60	7.80	0.87	23.19	7.92	0.88
Physical symptoms in the last week	5	1–5	15.59	5.36	0.83	13.34	4.93	0.81
Well-being in the last week	4	1–4	12.14	3.64	0.57	12.87	3.49	0.64
Emotional symptoms in the past six months	8	1–5	26.78	8.30	0.92	22.82	8.13	0.92
Physical symptoms in the past six months	5	1–5	15.63	5.719	0.87	13.00	5.18	0.87
Well-being in the last six months	4	1–4	12.42	3.91	0.76	12.84	3.81	0.76

Note. M = mean; SD = standard deviation; α = Cronbach’s alpha coefficient.

**Table 3 ejihpe-15-00133-t003:** Differences in intermediate determinants between participants with and without a mental health diagnosis (Mann–Whitney U Test).

Variables	Median	Median	*U*	*Z*	*p*	Effect Size (r)
	With diagnosis (n = 252)	Without diagnosis (n = 980)				
Well-being at home	524.48	640.16	100290.00	−4.631	<0.0001	0.132
Well-being in the neighborhood	552.46	632.97	107341.50	−3.211	<0.001	0.091
Well-being in religious organizations	542.09	635.63	104728.50	−3.903	<0.0001	0.111
Social welfare, culture and recreation	602.68	620.05	119996.50	−0.713	0.476	0.02
School wellness	605.85	619.24	120796.50	−0.535	0.592	0.015
Well-being at work	609.86	618.21	121807.00	−0.363	0.717	0.01
Expectation of social support	559.09	631.26	109013.50	−2.873	0.004	0.082
Healthy lifestyle habits	597.48	621.39	118687.50	−0.953	0.341	0.027
Psychosocial skills for life	552.00	633.09	107226.00	−3.230	<0.0001	0.092
Worry and emotional distress	700.73	594.84	102253.00	−4.231	<0.0001	0.121
Suicidal behavior	793.24	571.05	78942.00	−9.332	<0.001	0.266
Drug use	660.75	605.12	112329.50	−2.240	0.025	0.064
Conflicts due to drug use	643.90	609.45	116574.50	−1.416	0.157	0.04
Problematic internet use	622.01	615.08	122091.50	−0.278	0.781	0.008
Problems due to problematic internet use	644.53	609.29	116417.50	−1.441	0.150	0.041
Emotional symptoms in the last week	733.85	586.32	93907.00	−5.875	<0.0001	0.167
Physical symptoms in the last week	733.53	586.41	93989.00	−5.865	<0.0001	0.167
Well-being in the last week	549.61	633.70	106623.50	−3.360	<0.001	0.096
Emotional symptoms in the past six months	757.55	580.23	87936.50	−7.063	<0.0001	0.201
Physical symptoms in the past six months	750.78	581.97	89642.00	−6.732	<0.0001	0.192
Well-being in the last six months	580.78	625.69	114478.00	−1.796	0.073	0.051

Note. ***U*** = Mann–Whitney U statistic; ***Z*** = standardized test statistic; ***r*** = effect size calculated as ***Z***/√***N***.

**Table 4 ejihpe-15-00133-t004:** Results of the multiple regression predicting emotional symptoms and distress over the last six months without a diagnosis (N = 980).

Variables	B	95% CI (B)		SE B	t	β	R^2^	ΔR^2^
		LL	UL					
Step 1								
Constant	19.94 ***	12.719	27.169	3.682	5.417		0.13	0.13
Gender	−0.004	−0.996	0.989	0.506	−0.007	0.000		
Age	−0.026	−0.242	0.190	0.110	−0.236	−0.007		
Ethnicity	−0.773	−1.883	0.336	0.565	−1.368	−0.04		
Family Condition	−0.859	−3.246	1.529	1.217	−0.706	−0.02		
Marital status	3.758	−0.499	8.014	2.169	1.732	0.23		
Currently in a romantic relationship	−8.827	−17.374	−0.279	4.355	−2.027	−0.27 *		
Satisfaction with studies	−1.064	−1.515	−0.613	0.230	−4.629	−0.14 ***		
Works	−0.602	−1.755	0.552	0.588	−1.023	−0.03		
Dependent persons	0.725	−0.228	1.679	0.486	1.492	0.05		
Affiliated with health services	3.053	0.620	5.487	1.240	2.463	0.07 *		
Chronic health problem	−2.705	−11.778	6.368	4.623	−0.585	−0.10		
Treatment effectiveness: health problems	1.202	−2.164	4.569	1.715	0.701	0.12		
Functional diversity	0.484	−3.182	4.150	1.868	0.259	0.02		
Treatment or rehabilitation	0.277	−1.928	2.482	1.123	0.247	0.02		
Physical violence	4.332	2.776	5.888	0.793	5.464	0.17 ***		
Sexual violence	4.322	2.337	6.306	1.011	4.273	0.13 ***		
Psychological violence	5.304	3.944	6.664	0.693	7.654	0.24 ***		
Social network aggression	5.824	1.846	9.801	2.027	2.873	0.09 **		
Forced displacement	1.898	−0.698	4.494	1.323	1.435	0.04		
Step 2								
Constant	17.727 ***	10.150	25.305	3.861	4.591		0.29	0.16
Well-being at home	−0.109	−0.251	0.034	0.073	−1.491	−0.05		
Well-being in the neighborhood	0.026	−0.084	0.135	0.056	0.461	0.01		
Well-being in religious organizations	0.045	−0.020	0.110	0.033	1.356	0.04		
Social welfare, culture and recreation	−0.009	−0.069	0.051	0.031	−0.285	−0.009		
School wellness	−0.062	−0.205	0.080	0.073	−0.857	−0.03		
Well-being at work	0.056	−0.049	0.160	0.053	1.046	0.05		
Expectation of social support	0.012	−0.045	0.069	0.029	0.411	0.01		
Healthy lifestyle habits	−0.206	−0.297	−0.115	0.046	−4.453	−0.15 ***		
Psychosocial skills for life	0.007	−0.059	0.073	0.034	0.212	0.007		
Worry and emotional distress	0.476	0.337	0.616	0.071	6.694	0.21 ***		
Suicidal behavior	1.715	1.199	2.231	0.263	6.526	0.20 ***		
Drug use	0.207	0.002	0.412	0.104	1.983	0.06 *		
Conflicts due to drug use	−0.159	−0.319	0.001	0.082	−1.945	−0.07 *		
Problematic internet use	−0.107	−0.289	0.075	0.093	−1.157	−0.03		
Problems due to problematic internet use	0.424	0.238	0.611	0.095	4.470	0.15 ***		

Note. B = Unstandardized coefficient; CI = confidence interval; LL = lower limit; UL = upper limit; t = test of variance; SE-B = standard error of B; β = standardized coefficient. * *p* < 0.05. ** *p* < 0.01. *** *p* < 0.001.

## Data Availability

The research data of the article are available at https://osf.io/zwexn/?view_only=c94a7812de834057988c8923af0b7fc6 (accessed on 10 June 2025).

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
