# Peer review of "Exploring the Social Determinants of Mental Health in Colombian Young Adults"

_ejihpe, 2025, doi:10.3390/ejihpe15070133_

Round 1

Reviewer 1 Report

Comments and Suggestions for Authors

Thanks for the opportunity to review this valuable manuscript. The manuscript is generally well-written, methodologically sound, and practically important. I have several concerns below. I hope that the authors can find these suggestions useful if they would like to improve the manuscript.

(a) Clarification is required on how the convenience sampling of 1,232 undergraduates from a single city was mitigated to ensure external validity. Furthermore, outlining any stratification or weighting procedures and discussing limitations of generalisability would strengthen the Method and Discussion sections. 

(b) Further detail on the ethics procedures would be welcome: specify exactly how risk management for participants reporting active suicidal ideation was handled, and confirm whether a list of immediate support services was provided during online administration.

(c) The Social Determinants of Mental Health Questionnaire (SDMH) is appropriate, yet the paper relies mainly on internal consistency. In other words, a brief summary of prior construct-validity evidence, in addition to the indication of any measurement-invariance checks across sex or diagnostic status, would reassure readers of its robustness for subgroup comparisons.

(c) Most importantly, the diagnostic status (lifetime mental-health condition versus none) is typically confounded with socioeconomic disadvantage, neighbourhood safety, sex, and minority status; simple Mann–Whitney U comparisons, therefore, cannot establish whether observed differences are antecedents or consequences of psychopathology. Employing Propensity Score Matching (PSM) on sex, age, ethnicity, household income and community-safety indices, followed by post-match contrasts of key psychosocial variables, is now common in psychological and psychiatric research and is strongly recommended here.

(d) Self-reported lifetime diagnoses risk misclassification bias. A short paragraph acknowledging potential under- or over-reporting, and its possible impact on effect estimates, would balance the interpretation of group differences.

(e) The paper refers to “moderate to large” effect sizes for suicidal behaviour and emotional symptoms, yet Table 3 lists r values ≤ .27; revising the prose to align wording with conventional benchmarks will avoid inadvertent overstatement.

(f) The Discussion makes strong policy recommendations; tightening the link between regression findings and proposed multisectoral interventions (e.g., specifying how life-skills training might feasibly reduce violence exposure) would enhance translational value.

(g) The Data-Availability statement promises access “upon request.” Given the public-health importance of the dataset, the journal encourages authors to deposit de-identified data and syntax in a trusted repository, such as the Open Science Framework website, or to explain clearly any legal barriers to open sharing.

I hope that the authors can take every point of mine seriously so as to make their manuscript more publishable.  

Author Response

Dear reviewer 1, thank you very much for your observations, please see our responses in the attachment.

Reviewer 2 Report

Comments and Suggestions for Authors

Thank you for the opportunity to review "Exploring the Social Determinants of Mental Health Colombian Young Adults". This article was a quantitative investigation of the determinants of mental health problems and protective factors among university students in Colombia. The article, overall, is very well-written and clear. I offer the authors three methodological critiques and two grammatical edits. I will begin with the two methodological critiques:

  1. In the supplemental materials, the authors provide the internal consistency estimates for their measures, including Cronbach’s Alpha and McDonald’s Omega. While I approve of this addition, I believe the psychometrics of the scales are not fully reported. It is considered best practice to run confirmatory factor analyses (CFAs) on multi-item measures (especially with Likert-type items) to assess the quality of model fit to the observed data using common model fit indices (e.g., RMSEA ,CFI, TLI, SRMR, etc.; Hu & Bentler, 1999, Kline, 2023). It would be appropriate for the authors to add this psychometric information to the paper.
  2. The authors should speak to the potential Type-1 error inflation of running two separate hierarchical multiple regressions across two groups of participants (a total of four data runs). Other types of analyses, such as path analysis under the umbrella of structural equation modeling (SEM), would have allowed for one model to be run that captured all possible relations at once. For example, a multiple-groups analysis with mental health diagnosis as the grouping variable would have allowed for one full model with all relevant study variables to be examined at one time across both mental health diagnosis groups. I am not suggesting the authors need to do all this, but I am saying they should mention that inflation of Type-1 error was possible as a limitation and that alternative analytical choices are available for future studies.
  3. Related to the above, the authors note the unequal sample sizes and how the Mann-Whitney U test is sensitive to such differences. If this is the case, and the authors know it up front, why not do something else that is better equipped to handle the problem? I appreciate the acknowledgment of limitations, but it leaves the reader wondering why the authors didn’t do something about it when conducting the analyses in the first place.

Minor grammatical items:

Page 11, line 402: “However, the study by Kamis et al. (2023) highlights that the type and configuration of adversities is more important than *ir* total number, underscoring the need for targeted interventions tailored to specific adversity types.” It appears that the letters “ir” are included in error. This should be revised.

Page 12, line 463: The word “prioritized” should be present tense (“prioritize”).

That is all of my feedback. I applaud the authors on their insightful work and hope my comments are useful.

Author Response

Dear reviewer 2, thank you very much for your observations, please see our responses in the attachment.
